# A Comprehensive Review of Inorganic Sonosensitizers for Sonodynamic Therapy

**DOI:** 10.3390/ijms241512001

**Published:** 2023-07-26

**Authors:** Peng Chen, Ping Zhang, Navid Hussain Shah, Yanyan Cui, Yaling Wang

**Affiliations:** 1Beijing Engineering Research Center of Mixed Reality and Advanced Display, School of Optics and Photonics, Beijing Institute of Technology, Beijing 100081, China; 3220210447@bit.edu.cn (P.C.); 3220215086@bit.edu.cn (P.Z.); 3820192065@bit.edu.cn (N.H.S.); 2CAS Key Laboratory for Biomedical Effects of Nanomaterials and Nanosafety, National Center for Nanoscience and Technology of China, Beijing 100190, China

**Keywords:** sonodynamic therapy, SDT mechanisms, inorganic sonosensitizers, combination therapy

## Abstract

Sonodynamic therapy (SDT) is an emerging non-invasive cancer treatment method in the field of nanomedicine, which has the advantages of deep penetration, good therapeutic efficacy, and minimal damage to normal tissues. Sonosensitizers play a crucial role in the process of SDT, as their structure and properties directly determine the treatment outcome. Inorganic sonosensitizers, with their high stability and longer circulation time in the human body, have great potential in SDT. In this review, the possible mechanisms of SDT including the ultrasonic cavitation, reactive oxygen species generation, and activation of immunity are briefly discussed. Then, the latest research progress on inorganic sonosensitizers is systematically summarized. Subsequently, strategies for optimizing treatment efficacy are introduced, including combination therapy and image-guided therapy. The challenges and future prospects of sonodynamic therapy are discussed. It is hoped that this review will provide some guidance for the screening of inorganic sonosensitizers.

## 1. Introduction

Cancer poses a significant threat to human longevity that cannot be ignored [1]. The current treatments for cancer include surgical removal of the tumor, chemotherapy, radiotherapy, and immunotherapy to kill tumor cells and inhibit metastasis. However, these treatments often have severe side effects that result in great pain and economic burden for patients. In particular, surgical removal can lead to problems affecting normal human organs, along with complications after the operation [2]. In addition, simple surgery for small size tumors cannot completely eliminate, easy-to-exist residual tumor cells, thus surgical treatment is often combined with chemotherapy and radiotherapy treatment [3,4]. Unfortunately, chemotherapy and radiation do not just have bad effects on tumor cells, which often leads to damage to normal tissues and increased treatment resistance during treatment [5]. Similarly, immunotherapy-based treatments can lead to immune system damage due to an imbalance in the number of immune cytokines [6]. As a result, more research is focused on developing new strategies to reduce the side effects of treatments. One such strategy involves using materials that can kill cancer cells when exposed to light, known as photosensitizers [7,8]. These materials can generate reactive oxygen species (ROS) when exposed to specific light irradiation, while photothermal sensitizers can heat up under specific light irradiation to kill tumor cells. This has led to the development of photodynamic and photothermal therapies (PDT and PTT). However, although light therapy has overcome the side effects of traditional cancer therapy on the human body to some extent, light therapy has limited penetration effects on deep tissues, and its treatment scope may not include deep-seated tumors [9].

As a mechanical wave that differs from light waves, ultrasound has been widely used in both diagnosis and treatment in the medical field [10,11,12]. Sonodynamic therapy (SDT) was first discovered in 1989, when Umemura et al. observed that hematoporphyrin showed little cytotoxicity without external intervention but exhibited remarkable killing ability under ultrasound irradiation [13,14]. In comparison to traditional therapies and phototherapy, SDT offers several benefits, including the ability to penetrate deep tissues, control treatment time and space, and being non-invasive. These advantages make it an excellent choice for the treatment of deep tumors in the human body, and it has rapidly grown in biomedical applications. With the development of nanotechnology, sonosensitizers with enhanced imaging capabilities have been widely used to guide sonodynamic therapy (SDT) [15,16,17,18]. Compared to traditional imaging techniques, the use of sonosensitizers for molecular imaging offers advantages such as non-invasiveness, real-time monitoring, tracking, targeting, and high spatiotemporal resolution. These advantages provide a foundation for early diagnosis and visualization of pathological tissues during therapeutic processes. For most SDT processes, after entering the human body, acoustic sensitizers need to enrich in the tumor area and produce cytotoxic ROS under the action of ultrasound, so as to realize the killing effect on tumor cells. Importantly, ultrasound can penetrate deep tissues and focus precisely on the target tumor cells at different times, providing temporal and spatial control of treatment [19,20]. Although SDT and photodynamic therapy have some similarities in terms of their mechanism of action, they also have significant differences. Photodynamic therapy uses light to activate photosensitizers to generate ROS [21], whereas the killing of cancer cells by SDT may be related to the properties of the ultrasound itself. In general, the killing ability of SDT on tumor cells is mainly related to three factors: the frequency and power of ultrasound, the oxygen content of the tumor area, and the type of acoustic sensitizer. Among these, the sonosensitizer is the most crucial component [22].

Sonosensitizers are the most critical element of SDT, and research on their properties is a top priority. The ideal sonosensitizer must possess high acoustic sensitivity, specifically accumulate at the cancer site, be non-toxic without ultrasound, and rapidly metabolize from the body. The use of organic sonosensitizers has been extensively studied since 1989, and it remains a focus of current research. However, their limited retention in tumor areas has been observed, resulting in a limited therapeutic effect of SDT. With the continuous in-depth study of inorganic nanomaterials, inorganic sonosensitizers, such as TiO_2_ nanoparticles, MnO_2_ nanoparticles and super-small quantum dots (QDs), have been developed to improve chemical stability and reduce phototoxicity, providing a broad prospect for SDT [23].

Therefore, it is necessary to provide a comprehensive review of inorganic sonosensitizers, in order to inspire future screening and structural design of inorganic sonosensitizers. This review provides a comprehensive introduction to the possible therapeutic mechanisms of sonodynamic therapy and reviews the current application status and specific therapeutic mechanisms of inorganic nanomaterials in SDT, including metallic and non-metallic materials. Based on previous studies, methods to enhance the effectiveness of SDT are summarized, including sonodynamic combination therapy and image-guided therapy. Sonodynamic combination therapy refers to the integration of sonodynamic therapy with other existing therapies to complement their respective limitations and achieve a synergistic effect. Simultaneously, through the design of inorganic sonosensitizer structures, it can serve as an integrated platform for diagnosis and treatment, enabling image-guided therapy. Finally, the remaining challenges and key issues in this field are discussed to promote the future development of SDT. The main section of the article is illustrated in Figure 1.

## 2. Possible Mechanisms for SDT

Ultrasound has the ability to penetrate deep into tissue, allowing for the activation of sonosensitizers at a specific location and leading to cell damage or the disruption of the redox balance, ultimately resulting in cell apoptosis. Despite the potential benefits of SDT, the exact mechanism underlying its effectiveness remains unclear. However, it is believed that the possible mechanisms of SDT may include ultrasonic cavitation, ROS generation, and activation of immunity (Figure 2a).

### 2.1. Ultrasonic Cavitation

Cavitation is the interaction between a sound field and a bubble, which can be classified as either inertial or non-inertial cavitation, with different effects in the sonodynamic therapy process. Inertial cavitation can cause tissue damage by generating high temperature and pressure. Non-inertial cavitation generates heat, microflow, and local shear stress due to bubble oscillations [24,25]. During inertial cavitation, the cavitation volume steadily increases to the resonance volume, which triggers an explosion [26]. Ultrasonic pressures that reach or exceed a threshold can cause a microbubble to burst immediately. The shock wave generated during the rupture will form a steep rise in temperature and pressure in a very small space, causing damage to the membrane structure and skeleton of the cell, thus causing a killing effect on the cell [27]. The mechanical shear force from non-inertial cavitation may also contribute to tumor cell death by damaging cell membranes [28].

SDT has been found to have the potential to disrupt the blood–tumor barrier and impede the formation of vascular lumen, resulting in a reduction in blood flow and energy supply to inhibit tumor growth [29]. In a study by Ho et al. [30], silica nanoparticles were utilized as an acoustic sensitizing agent to deliver the anti-tumor drug doxorubicin. Immunofluorescence staining revealed that all blood vessels with a diameter less than 10 μm were ruptured after ultrasound exposure, while for vessels with diameters between 10 μm and 40 μm, the proportion of rupture was 57%. The destruction of blood vessels by ultrasonic cavitation allowed for increased drug delivery and retention in the tumor area, with the amount of drug in cells subjected to ultrasound being 2.98 times higher compared to cells without ultrasound. These findings suggest that ultrasonic cavitation can effectively promote necrosis of the target tissue and increase the amount of sonosensitizer released in the target area by inducing the rupture of blood vessels.

Cavitation caused by ultrasound is one of the important mechanisms of killing tumor cells. In addition to the inhibitory and destructive action of cavitation itself on tumor growth, it is also an important cause of ROS generation in the process of SDT, which will be detailed in the next part.

### 2.2. ROS Generation

Reactive oxygen species (ROS) are toxic byproducts of aerobic respiration that can damage DNA, proteins, and lipids. Similar to PDT, the mechanism of ROS in SDT disrupts the redox balance of tumor cells, resulting in cell death. This is currently the most widely recognized mechanism of SDT. Figure 2b shows the process of ROS generation and how they cause mitochondrial damage.

Under ultrasonic irradiation, the sonosensitizers absorb the energy carried by ultrasound to generate ROS, including singlet oxygen (^1^O_2_), hydroxyl radical (•OH), and superoxide anion (•O_2_^−^) [31]. During the process of SDT, sonosensitizer structures are excited from the ground state to a higher energy state by absorbing ultrasound energy. When the sonosensitizer transitions from the higher energy state to the lower energy state, a large amount of energy is released to the adsorbed molecules, accompanied by the generation of ROS [32]. The ROS produced can effectively damage intracellular DNA and proteins, promote cellular lipid peroxidation, and induce apoptosis of target cells to achieve SDT [33].

Currently, cavitation is considered the primary process for generating ROS by sonosensitizers under ultrasound exposure. Moreover, pyrolytic effects and sonoluminescence (SL) are also responsible for ROS generation [34]. Firstly, the sonosensitizer promotes cavitation, and ultrasonic cavitation in turn activates the sonosensitizer, generating two types of ROS through different reactions. (i) Excited sonosensitizers can react directly with dissolved oxygen molecules or other substrate molecules to generate free radicals. (ii) By transferring the energy released from the sonosensitizers back to the oxygen, highly active singlet oxygen(^1^O_2_) can be produced, which was confirmed by experimental results to be the most ROS produced in SDT [35]. A high concentration of singlet oxygen can irreversibly damage tumor cells by destroying membranes, contracting the cytoskeleton, and disrupting mitochondrial membrane potential. Another pathway for ROS generation based on the “pyrolysis effect” has been proposed. The local high temperature generated by inertial cavitation can directly decompose H_2_O to generate ROS through a series of processes, causing damage to tumors [36]. Additionally, sonoluminescence (SL) can activate the sonosensitizer to produce ROS. During ultrasonic cavitation, local high temperature and pressure are generated, accompanied by the release of light, known as the sonoluminescence phenomenon [37]. Sonoluminescence is the phenomenon in which sound energy is transformed into light energy, typically in the form of subnanosecond pulses with a wavelength range of 200–800 nm. When a sonosensitizer is excited by a specific wavelength of light, similar to photodynamic therapy, ROS are generated. The excited state of the sonosensitizer transfers energy to surrounding oxygen, resulting in oxidative reactions with adjacent tumor cells that have cytotoxic effects, ultimately leading to cell damage and death. Although experimental evidence supports the aforementioned ROS generation pathways, it remains unclear which pathway is the primary mechanism for ROS production in SDT. This remains a critical area of research, as a more comprehensive understanding of the ROS production pathway could lead to the development of more effective sonosensitizers.

### 2.3. Activation of Immunity

The immunotherapy process induced by SDT is illustrated in Figure 2c. T lymphocytes, dendritic cells, macrophages, and other cells contribute to antitumor immunity through cellular immunity. Studies have demonstrated that SDT can promote the activation and proliferation of immune cells, primarily in three ways. Firstly, SDT treatment can enhance the activation of T cells and the antigen presentation process to better recognize and eliminate cancer cells [38]. Secondly, SDT can induce the upregulation of DCs and enhance the anti-tumor immune response [39]. Finally, tumor debris generated by SDT can activate immune and anti-inflammatory responses, thereby promoting the transformation of bypass-activated macrophage M2 into macrophage M1 and enhancing the immune response [40]. The primary effect of these mechanisms is to boost the patient’s immunity, which helps to fight the cancerous cells.

Through the induction of an immune response, SDT not only activates and proliferates immune cells but also upregulates cytokines to strengthen the body’s anti-tumor immunity. Research has confirmed that during SDT treatment, ROS are produced in high quantities to destroy the mitochondrial membrane, reducing the membrane potential in mitochondria. Additionally, ROS mediates apoptosis through the mitochondrial caspase pathway, releasing tumor antigens and promoting CTL infiltration [41]. The SDT process can also increase the expression of calreticulin (CRT) on the surface of tumor cells, stimulate the release of cytokines, and promote a specific anti-tumor immune response [42].

Studies have shown that the SDT process enhances the immune response against tumors by inducing immunogenic cell death (ICD) and releasing tumor-associated antigens. ICD is a result of damage-associated molecular patterns (DAMPs), which are a series of highly immune-stimulating signals. These DAMPs attach to their respective receptors, activating immune cells. For instance, extracellular ATP produces “find me” signals that stimulate the rapid aggregation of antigen-presenting cells (APCs) around apoptotic tumor cells. HMGB1 binds to Toll-like receptor 4, which helps APCs present tumor antigens to T cells. CRT is transported to the surface of tumor cells, generating “eat me” phagocytosis signals that prompt phagocytes, such as dendritic cells, to recognize and phagocytose cancer cells. The ICD phenomenon, induced by SDT and derived tumor cell lysates, can trigger a strong immune response.

### 2.4. Discussion

Although there are other hypotheses regarding the mechanism of SDT, its efficacy in killing tumors is undeniable. Based on widely accepted theories, SDT generates ROS by utilizing ultrasound as an excitation source, resulting in apoptosis through cavitation, sonoluminescence, or pyrolysis processes. Cell necrosis can also occur due to mechanical and thermal damage resulting from cavitation.

It is worth noting that during the process of SDT, ultrasound can also cause a temperature increase in the focal region, although the magnitude of this increase is relatively small [43]. Unlike high-intensity focused ultrasound (HIFU), the cytotoxic effect of SDT on cancer cells primarily arises from the generation of reactive oxygen species (ROS) following the absorption of ultrasound energy by sonosensitizers, rather than the local hyperthermia induced by ultrasound. Although the temperature elevation caused by ultrasound during SDT does not directly kill tumor cells, we believe that by rationally designing the structure of the sonosensitizer, it is possible to utilize temperature changes to achieve spatiotemporal control of structural transformation of the sonosensitizer and drug release.

## 3. Inorganic Nanomaterials Applied in Sonodynamic Therapy

There are currently two types of sonosensitizers available: organic and inorganic. In comparison to organic materials, inorganic materials exhibit fast metabolism, low toxicity, and good stability. Inorganic micro/nanomaterials offer a novel platform for sonodynamic therapy due to their unique structure, composition, and multifunctionality. Notably, certain small-sized inorganic nanoparticles can infiltrate blood vessels via tissue extravasation and penetrate into tumors [44,45]. The application of inorganic nanomaterials in acoustic sensitization technology mainly includes the use of sonosensitizers or acoustic sensitization carrier. Numerous studies have explored the use of inorganic materials as carriers for organic sonosensitizers [46,47,48,49]. This review aims to provide a summary of the application of inorganic materials as sonosensitizers, with a brief introduction to the use of inorganic materials as carriers of sonosensitizers. Based on the main components present in the nanostructure, sonosensitizers can be categorized into two types: metal-based and non-metallic.

### 3.1. Metal-Based Sonosensitizers

#### 3.1.1. TiO_2_ Nanostructures-Based Sonosensitizers

TiO_2_ nanoparticles are widely utilized in various fields such as food, cosmetics, energy, and environmental protection due to their semiconducting properties. TiO_2_ nanoparticles are favored over other materials because of their narrow band gap (3.2 eV) and excellent biocompatibility, which make them highly suitable for various applications. Under UV irradiation, the electrons in the valence band of TiO_2_ nanoparticles can be excited to move to the conduction band, and these electrons can react with the surrounding oxidizing substances (O_2_ and H_2_O) to produce ROS, such as •OH, O_2_^−^, H_2_O_2_ or other active substances. Consequently, TiO_2_ nanoparticles have been utilized for tumor photodynamic therapy [50]. TiO_2_ nanoparticles have limited tissue penetration when activated by UV irradiation, and the recombination of electrons and holes is easy, which limits their use in photodynamic therapy. However, Harada et al. [51] discovered that ultrasonic-activated TiO_2_ nanoparticles could effectively kill tumor cells, and subsequent studies have extensively investigated TiO_2_ nanoparticles as a representative sonosensitizer [52]. As one of the widely used photosensitizers, TiO_2_ nanoparticles are basically non-toxic to experimental cells or animals, and they have great potential as sonosensitizers.

However, bare TiO_2_ nanoparticles are known to be unstable, and they cannot effectively accumulate in tumor areas due to easy capture by the reticuloendothelial system. This shortcoming hinders ideal therapeutic efficiency. Surface modification using high-polymer materials such as proteins, glucans, and polyionic complexes can improve this issue. You et al. [53] prepared hydrophilic TiO_2_ nanoparticles for cancer therapy (HTiO_2_ nanoparticles) by coating TiO_2_ nanoparticles with carboxymethyl glucan (Figure 3a). Compared with bare TiO_2_ nanoparticles, long-period HTiO_2_ nanoparticles have better therapeutic effect. TiO_2_ nanoparticles were also grafted with polyallylamine to form core-shell polyionic composite micelles (Figure 3b) resulting in TiO_2_ nanoparticles with high dispersion stability under physiological conditions [54]. The fluorescence labeling experiment confirmed the ideal speed of tumor aggregation and cell entry of TiO_2_ nanoparticles. Under US irradiation, the nanoparticles produce ROS, which was confirmed by the reactive oxygen species staining experiment as ^1^O_2_, and the fluorescence signal in the ultrasonic region was strong. Shimizu et al. enriched TiO_2_ nanoparticles in tumor tissues by wrapping them in different proteins [55,56,57]. The results confirm that protein-modified TiO_2_ nanoparticles greatly improve the enrichment of nanoparticles in tumor regions and targeting of different proteins to different tumor cells is also evident. Therefore, the excellent performance of TiO_2_ enables efficient SDT, and its instability can be avoided by simple encapsulation. To improve the stability and tumor targeting of TiO_2_ nanoparticles, biomimetic cell membranes-based nano-composite structures have been applied. By coating the cancer cell membrane (CM) on the surface of nanoparticles, Ning et al. [58] prepared C-TiO_2_/TPZ@CM composite sonosensitizer (Figure 3c). As a result of the presence of CM, tumors were targeted through homologous binding.

Although TiO_2_ nanoparticles with high stability can be obtained through surface modification, single TiO_2_ sonosensitizers still have certain limitations due to rapid recombination of electrons and holes, leading to reduced ROS production efficiency. To overcome this challenge, a strategy that involves doping noble metal nanoparticles into TiO_2_ nanomaterials has been developed to prevent electron-hole recombination, and thereby increase the ROS generation efficiency. This approach is commonly used with precious metal nanoparticles (such as gold and platinum) and semiconductors. Perota et al. [59] developed Au/TiO_2_ nanocomposite applied to PTT and SDT for the synergic treatment of melanoma in vitro. When active US energy is applied, charge separation occurs in TiO_2_, and the valence band electrons escape the positive holes. The lower fermi level of Au allows it to act as a recipient of transition electrons, while TiO_2_ retains its holes. This process effectively suppresses electron-hole recombination, and extends the lifetime of electrons, resulting in improved ROS generation efficiency. The results of the experiments indicated that treatment led to a substantial increase in apoptosis of melanoma cell lines. However, due to the high cost of noble metal nanoparticles, doping with other low-cost elements is an effective approach to enhance ROS production. The nanomaterials prepared by doping tungsten into TiO_2_ nanorods (W-TiO_2_) have good tumor eradication ability in the combination of acoustic and chemodynamic therapy for human osteosarcoma [60]. The synthesis of multifunctional W-doped TiO_2_ (W-TiO_2_) nanorods are shown in Figure 4a. PEG-modified W-TiO_2_ ultrafine nanorods with high dispersion and excellent biocompatibility were synthesized organically in a high-temperature environment. Experimental results in Figure 4b–e shows that, due to W-doped TiO_2_ nanorods having a smaller band gap (2.3 eV), they are more efficient under ultrasonic irradiation than undoped TiO_2_ (3.2 eV) (Figure 4f). As the band gap of composite materials is reduced, the ROS generation efficiency is improved. Here, the band gap reduction can be attributed to the energy state of the W 5d level located below the TiO_2_ conduction band. In addition, W-doping in TiO_2_ nanorods can not only catalyze the generation of •OH from overexpressed H_2_O_2_, but also convert glutathione into glutathione disulfide (GSSG), which enhances oxidative stress and SDT in tumor cells’ TME. The schematic diagram of the ROS generation pathway enhanced by W-TiO_2_ nanoparticles is shown in Figure 4g. CDT + SDT inhibited osteosarcoma growth completely (Figure 4h), while SDT alone inhibited only partial growth (52% inhibition rate). The results show that the doping of tungsten plays an important role in enhancing the SDT effect of TiO_2_ nanoparticles. Carbon structures are frequently utilized to modify metal oxides and provide a protective carbon layer around the metal oxide particles. Cao et al. [61] prepared lamellar TiO_2_/C NPs for SDT, which achieved good results in the treatment of pancreatic cancer. In the composite structure, supported by MOF-derived carbon structures, TiO_2_/C remained stable even after repeated US stimulation, whereas ROS generation was improved. The antitumor activity of nanoparticles in vitro and in vivo was verified to inhibit tumor growth and induce DNA damage in cancer cells.

Defect-rich sonosensitizers can enhance the efficacy of sonodynamic therapy by inhibiting electron-hole pairing. Inspired by traditional photocatalysis, Han et al. [62] constructed a core/shell structured TiO_2_@TiO_2−x_ nanomaterials. Under ultrasonic irradiation, the oxygen defect in TiO_2−x_ promotes and improves the separation of electron and hole, which significantly improves efficacy of acoustic triggered tumor therapy. To further enhance charge transfer and the ability to regulate the tumor microenvironment, Geng et al. [63] designed Cu_2−x_O@TiO_2−y_ Z-scheme heterojunctions for combined SDT and CDT. Using Cu_2−x_O nanodots and TiO_2−y_ nanosheets with narrow band gaps, and Z-scheme junctions formed between their interface on the nanoscale, would greatly improve SDT. In addition, the nanocomplex has excellent redox activity and can regulate TME by Cu^+^/Ti^3+^ catalyzing endogenous H_2_O_2_ and Cu^2+^-mediated endogenous GSH depletion to produce •OH. The composite NPs exhibit 100% tumor inhibitory rate and better survival when combined with chemodynamic therapy and sonodynamic therapy.

#### 3.1.2. Manganese-Based Composite Materials

In tumor microenvironments (TMEs), common characteristics such as low pH, high concentrations of glutathione (GSH), excess production of hydrogen peroxide (H_2_O_2_), and hypoxia create a suitable environment for tumor cell growth and promote tumor progression, metastasis, and drug resistance [64]. As a result, the development of intelligent nano-therapy systems that consider the unique environmental features of TMEs holds great promise in improving the efficacy of cancer treatment. Manganese-based composite materials, due to their tunable structure/morphology, pH-responsive degradation, and excellent catalytic activity, have gained increasing attention as an excellent TME-responsive treatment platform in SDT [65].

As an ideal biomaterial responsive to TME, MnO_2_ can regulate the TME by converting to oxygen and Mn^2+^ at an acidic pH, which triggers the breakdown of endogenous tumor H_2_O_2_, glucose consumption, and oxidation of glutathione (GSH) to oxidized glutathione (GSSG) [66]. This process catalyzes O_2_ formation from the high H_2_O_2_ concentration in the TME, which alleviates tumor-related hypoxia. Specifically, glucose oxidase converts glucose into gluconic acid and H_2_O_2_, while MnO_2_ catalyzes H_2_O_2_ into O_2_. Liu et al. constructed the ultrathin-FeOOH-coating MnO_2_ nanopheres (MO@FHO) as a high efficiency sonosensitizer that not only improves reactive oxygen species production, but also regulates tumor microenvironments from multiple aspects to achieve efficient SDT (Figure 5a) [67]. The properties of the composite can be observed in Figure 5b, which shows a smaller band gap compared to pure MnO_2_. MnO_2_ is utilized as the active core for generating ultrasonic-initiated reactive oxygen species. By combining FeOOH and MnO_2_, FeOOH can act as a hole conductor that facilitates the separation of electron-hole pairs triggered by ultrasound, accelerating the generation of reactive oxygen species. In addition, SDT can also be enhanced by adjusting TEM. The catalase-like characteristics of MnO2 core and the cocatalyst function of the covered FeOOH can effectively alleviate tumor hypoxia by decomposing endogenous H_2_O_2_ into O_2_ (Figure 5c). Apart from MnO_2_, the unique properties of polyvalent manganese oxide (MnO_x_) also show extensive potential for SDT. MnO_x_ can produce ^1^O_2_ under acidic conditions, which has the ability to kill cancer cells. Good SDT results can be achieved by targeting TEM with acidic properties [68]. Oxygen vacancies (OVs) in semiconductor metal oxides play a vital role in regulating many reactive oxygen species. Studies have indicated that OVs in metal oxide semiconductors significantly influence reactive oxygen species production. Sun et al. [69] developed a sonosensitizer that is not affected by low oxygen environment through oxygen vacancy engineering strategy, namely iron-doped multivalent manganese oxide nanoparticles (FDMN). The number of oxygen vacancies in the synthesized nanoparticles can be regulated by using the strategy of constructing vacancy engineering. The researchers doped iron ions (Fe^3+^) into MnO_x_ nanoparticles, which have a similar atomic radius to Mn4+, thus forming iron-doped composites. After pegylation, the synthesized FDMNs exhibit uniform morphology, ultra-small size, and excellent water solubility. FDMN with narrow band gap can be used as an effective sonosensitizer to achieve efficient ROS production, and their ROS quantum yield is better than that of MnO_x_ NP due to the rich OVs structure. Moreover, the composite material possesses an abundance of oxygen vacancies that can adsorb a large number of oxygen molecules, thereby overcoming the hypoxic conditions of the TME and enabling efficient ROS production under ultrasound stimulation.

Moreover, ultra-small MnWO_x_-PEG nanoparticles have been identified as a novel sonosensitizer for multimodal imaging-guided sensing and diagnostic techniques [70]. The synthesized MnWO_x_ nanoparticles were then functionalized using the amphiphilic polymer polyethylene glycol grafted poly (C_18_PMH-PEG) to improve their water suspension stability and biocompatibility. The special structure of the composite material can facilitate the separation of e^−^ and h^+^ by creating additional sites to capture electrons. In addition, MnWO_x_-PEG nanoparticles prevented scavenging of reactive oxygen species by glutathione and further enhanced the effect of SDT treatment (Figure 5d). Furthermore, Mn^2+^ has been reported to have the ability to regulate mitochondrial function and induce apoptosis [71]. Zhang et al. developed a novel sonosensitizer, manganese carbonate nanoparticles (MnCO_3_ NPs), by the inverse microemulsion method, for enhanced SDT [72]. MnCO_3_ nanoparticles can efficiently produce hydroxyl radical (•OH) and singlet oxygen (^1^O_2_) under ultrasonic radiation. In addition, MnCO_3_ nanoparticles will release CO_2_ and Mn^2+^ due to degradation caused by the local acidic microenvironment. The resulting CO_2_ bubble can be stimulated by ultrasound to explode, resulting in irreversible cell death. The intracellular treatment process is illustrated in Figure 5e. Additionally, MnCO_3_ nanoparticles exhibited good ultrasound imaging contrast ability in SDT guidance, attributed to the CO_2_ release.

**Figure 5 ijms-24-12001-f005:**
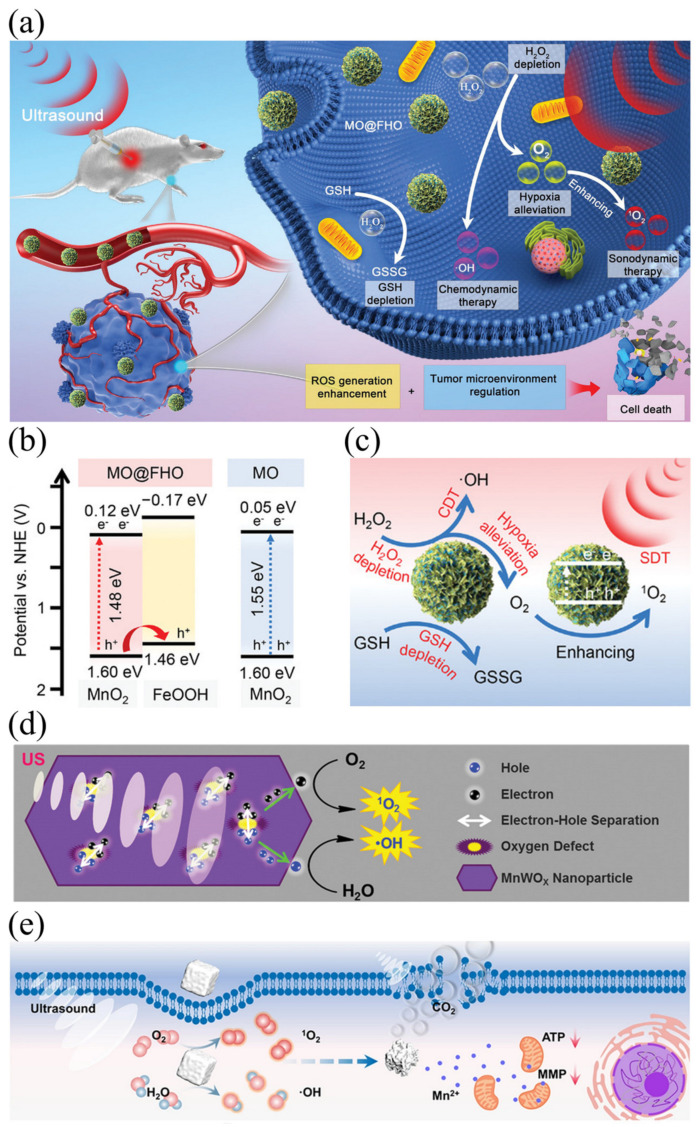
(**a**) Schematic diagram for the anticancer mechanism of MO@FHO; (**b**) the energy band diagrams MO and MO@FHO; (**c**) the overall scheme showing mechanisms of MO@FHO sonosensitizer via synergistic effects to achieve the tumor therapy [67]; (**d**) the proposed mechanism of ROS generation by MnWO_x_−PEG under ultrasound-irradiation [70]; (**e**) schematic illustration of intracellular treatment of MnCO_3_ NPs [72].

#### 3.1.3. Other Metal-Based Sonosensitizers

Tumor starvation therapy is a strategy to suppress tumor growth by depleting glucose in the tumor. Based on the principle of tumor starvation and acoustic dynamic therapy, Zhou et al. [73] designed and synthesized a novel bimetallic PdPt-based sonosensitizer that enhanced the therapeutic effect by using photothermal effects. GO_x_ was compounded on the surface of PdPt NPs, and IR780 was coated around the NPs by strong electrostatic action. After entering cancer cells, it depletes intratumoral glucose through a GO_x_-mediated tumor starvation catalytic reaction, and subsequently establishes a H_2_O_2_-rich microenvironment. Furthermore, the reactive oxygen species produced by ultrasound disrupted the intracellular redox balance and led to apoptosis. Meanwhile, PdPt nanoparticles were capable of decomposing hydrogen peroxide into oxygen. The generated oxygen not only accelerated glucose consumption to promote the effect of hunger treatment, but also facilitated the SDT process. Additionally, the composite nanoparticles exhibited significant photothermal response to near-infrared light, and the photothermal effect produced by the composite nanoparticles could synergize with SDT treatment and enable photoacoustic (PA) in-situ imaging.

It is noteworthy that the cavitation effect induced by ultrasound can lead to the local production of high temperature, high pressure, and acoustic luminescence [74]. However, the high pressure (≈1 × 10^8^ Pa) generated by bubble collapse is often overlooked. Based on the aforementioned analysis, piezoelectric materials have the potential to facilitate SDT by utilizing the transient high pressure generated by the cavitation effect. Piezoelectric materials are a type of dielectric material with an asymmetric crystal structure, including ZnO, perovskite structure materials, two-dimensional ultrathin materials, and layered bismuth-based materials [75]. When subjected to ultrasound, piezoelectric materials become polarized by sound waves, creating an internal electric field (IEF). The generation and retention of IEF is an effective method to promote charge separation [76]. However, their application in SDT is still in its nascent stage, and few studies have been conducted in this area. Dong et al. presented a successful example of the use of piezoelectric sonosensitizers for endogenous sonodynamic enhancement therapy [77]. Here, ultra-thin 2D Bi_2_MoO_6_-poly(ethylene glycol) nanoribbons (BMO-NRs) were designed as a piezoelectric sonosensitizer for glutathione (GSH)-enhanced SDT, which is activated by endogenous GSH and amplified by exogenous ultrasound. Figure 6a presents a schematic diagram of two-dimensional piezoelectric Bi_2_MoO_6_ sonosensitizer SDT. In vitro cell experiments showed strong green fluorescence under the condition of BMO+US, indicating high levels of ROS (Figure 6b). In vivo experiments on mice showed significant tumor suppression (Figure 6c–e). This indicates that BMO NRs effectively inhibit tumor growth under US irradiation and, at the same time, shows the great potential of piezoelectric materials as sonosensitizers in SDT. After entering cancer cells, nanoparticles can help disrupt cell homeostasis by consuming GSH, and GBMO produced under the action of GSH has more OVs. The presence of these OVs can capture electron hole pairs generated under ultrasonic excitation, thus achieving high levels of SDT by increasing the production of reactive oxygen species.

In addition, certain transition metal oxides also possess significant potential for achieving SDT. Due to its superparamagnetic properties, Fenton-like reactivity, and mimicking activity of catalase, Fe_3_O_4_ has garnered increasing attention in the field of SDT [78]. Yeh et al. [79] designed Fe_3_O_4_-loaded sonosensitizers for SDT by utilizing the stable properties of poly(lactic-co-glycolic acid) (PLGA) polymer in various solutions, as well as its complete decomposition under ultrasound. The hydrophilic core of the PLGA polymer was used to encapsulate H_2_O_2_ through a double emulsion process, while Fe_3_O_4_ was loaded on the hydrophobic PLGA shell. This structural design allows for the release of Fe_3_O_4_ and H_2_O_2_ under the influence of ultrasound, leading to the generation of ROS and O_2_, thus achieving the inhibition of tumor growth. ZnO has become a research hotspot in the field of SDT due to its semiconductor properties and excellent biocompatibility. For example, Zhang et al. [80] reported the utilization of gadolinium (Gd)-doped oxygen-deficient zinc oxide nanoparticles (D-ZnOx:Gd) for cancer treatment under MRI imaging guidance. In addition to facilitating the separation of electron-hole pairs, the oxygen defects on the nanobullets also facilitate the adsorption of H_2_O and O_2_ molecules, thereby enhancing the generation of ROS. This suggests that ZnO may be a promising inorganic sonosensitizer. The efficacy of Al_2_O_3_ particles in inducing cytotoxicity in tumor cells under ultrasound has also been demonstrated [81]. They found that ultrasound stimulation is a necessary condition for the induction of cell apoptosis by micro-sized Al_2_O_3_ particles, and the concentration and size of the particles significantly affect the effectiveness of sonodynamic therapy (SDT).

### 3.2. Non-Metallic Sonosensitizers

#### 3.2.1. Silicon-Based Sonosensitizers

Besides metal-based sonosensitizers, silicon nanostructures are also extensively employed in SDT [82,83]. As inorganic semiconductor materials, silicon nanowires (SiNWs) have been widely used in electrocatalysts, nanodevices, and photovoltaics. It is found that its properties under ultrasound can also be used in sonodynamic therapy. Osminkina et al. [17] fabricated silicon nanomaterials (silicon nanowires, about 5 μm long, 200 nm inner diameter). These nanoparticles showed good biocompatibility, that is, cell viability was maintained at 95% even when incubated with silicon nanoparticles at a high concentration of 1 mg/mL. In contrast, treatment of cells with SI-NP and ultrasound resulted in a decrease in the survival rate to approximately 50%, while ultrasound alone had no significant effect on the survival rate. These findings suggest that SI-NPs have low cytotoxicity and can be used as an effective sonosensitizer for cancer sonodynamic therapy. The mechanism underlying this treatment can be attributed to the direct absorption of energy released by ultrasound and the indirect absorption of high temperatures caused by the cavitation effect. Furthermore, Sun et al. [84] successfully modified silicon nanowires with platinum nanomaterials for photothermal enhanced SDT (Figure 7a). The Si-Pt nanocomposites were synthesized by modifying Pt nanoparticles on SiNWs using in situ reduction. Under ultrasonic treatment, they generate abundant reactive oxygen species that act as an effective sonosensitizer for SDT. In tumor microenvironments, Si-Pt NCs exhibit good chemokinetic therapeutic activity by converting excess hydrogen peroxide into ROS. Due to the uniform size and distribution of Pt NPs, the chemodynamic therapy and sonodynamic therapy effects are stronger than those of pure Pt NPs. As shown in Figure 7b, by using DPBF as a probe for ^1^O_2_, it can be seen that the composite +US group produced the most ROS. Additionally, Si-Pt nanocrystals exhibit a photothermal effect, which may increase the combined effects of SDT and CDT. This effect is evident in the infrared image after laser irradiation (Figure 7c). Moreover, DCFH-DA staining of tumor sections (Figure 7d) 24 h after treatment confirmed the production of abundant ROS in tumor cells. Both in vivo and in vitro experiments have confirmed that the nanocomposite can significantly inhibit the growth of tumor cells and can be used for combination therapy involving SDT + CDT.

The use of mesoporous silicon in biomedicine, particularly in drug delivery, has gained wide acceptance. Its advantages, such as easy preparation, large loading area, good biocompatibility, surface modification, and tumor accumulation, make it an attractive option. In fact, studies have shown that nanoparticles containing mesoporous silicon are effective carriers of tumor sonosensitizers [85]. Mesoporous silica nanoparticles (MSNs) are also effective drug carriers with high loading capacity and biocompatibility [86]. Previous studies have shown that MSN surfaces can be modified to be hydrophilic or hydrophobic, resulting in air nanobubble accumulation at the interface between them [87]. Interfacial nanobubbles (INBs) can be trapped in the mesoporous cavity of a hydrophobic MSN and remain stable for at least four days. Under the effect of ultrasound, in addition to destroying blood vessels, these INBs produce ROS, which damage cells as well [88]. The first ROS produced are ·H and ·OH, which are generated by the pyrolysis of water during cavitation. In turn, H_2_O_2_, peroxyl radicals, and other reactive oxygen species are formed to kill tumor cells.

#### 3.2.2. Carbon-Based Sonosensitizers

It has been over 30 years since carbon nanomaterials gained significant attention. Among the most common materials are graphene, fullerenes, and carbon dots, all of which are characterized by a hexagonal lattice arrangement of sp2 hybrid carbon atoms. These materials exhibit unique chemical and physical properties that make them useful in a wide range of applications. Graphene-based materials possess unique physical and chemical properties that make them promising for biomedical applications, particularly in cancer therapy [89,90]. More recently, graphene-based materials have also been combined with sensitizers that have cytotoxic effects on external triggers such as light or ultrasound for tumor-specific therapy. Sonodynamic therapy has shown superior therapeutic potential compared to conventional photodynamic therapy, with high tissue penetration when combined with graphene-based materials. Lee et al. [91] reported a therapeutic strategy to target metastatic ovarian cancer spherules using graphene nanoribbons (GNR) functionalized with four-armed polyethylene glycol (PEG) and sonosensitizer chlorine e6 (Ce6). Compared with GO-PEG, GNR functionalized with four-arm polyethylene glycol (GNR-PEG) showed enhanced cytocompatibility and better tumor sphere adhesion blocking effect on healthy mesodermal cells. Importantly, GNR-PEG provided a more durable adhesion blocking effect compared with conventional antibody blocking methods. In addition, GNR-PEG loaded with the sonosensitizer chlorine e6 (GNR-PEG-Ce6) was able to kill ovarian cancer spheres by sonodynamic therapy. The rapid recombination of sonically excited electron hole pairs in graphene materials greatly limits its SDT applications. Wang et al. [92] proposed a highly efficient sonosensitizer system for enhanced sonodynamic therapy constructed on reduced graphene oxide (rGO) nanosheets, bridging zinc oxide and gold nanoparticles, and coated with polyvinylpyrrolidone (PVP). Under ultrasound irradiation, ZnO nanoparticles generate separated electron-hole pairs, and the narrow band gap of rGO nanosheets facilitates electron transfer from ZnO to Au nanoparticles, which efficiently restrains the recombination of the electron-hole pairs, significantly augmenting the production of ROS and thus killing cancer cells.

Functional fullerenes, including polyhydroxy fullerenes (PHFs), have shown promise in the diagnosis and treatment of tumors. Yumita et al. [93] demonstrated that the antitumor effect of PHFs induced by ultrasound was caused by the ROS produced. The closed spherical carbon nanostructure C_60_ fullerene (C_60_) is used in a variety of biomedical applications because its unique structure can elicit antiviral, antimicrobial, and anticancer activities [94]. Radivoievych et al. [95] investigated the effect of low-intensity ultrasound combined with C_60_ on human cervical cancer HeLa cells and observed its potential value in the sonodynamic treatment of cancer cells.

Carbon dots (CDs) have attracted much attention due to their unique characteristics, including low synthesis cost, good biocompatibility, high photostability, strong catalytic activity, and flexible surface function [96,97]. Ren et al. confirmed that CDs has the possibility to generate ROS under ultrasound and thus be applied to SDT, and studied the generation pathway of ROS [98]. They also studied the pathway of ROS generation and found that ultrasound irradiation can transform the oxygen-containing groups on the surface of CDs into ROS. These transformation reactions are reversible and can be controlled by the pH of the aqueous solution, providing the possibility of controlling the ROS yield in the microenvironment. Compared with typical TiO_2_ sonosensitizers, CDs showed nearly twice the ROS generation efficiency under the same conditions due to the presence of surface groups.

In Summary Inorganic nanomaterials, particularly TiO_2_, have great potential for SDT applications due to their unique physicochemical properties and biological safety. Surface modification and structural improvements can further enhance their efficiency. Nanoparticles with oxygen defects exhibit additional properties, such as nano-enzyme activity, besides promoting ROS production. While combining different inorganic materials provides advantages, certain challenges like biodegradability, biocompatibility, and solubility remain. A possible solution could be combining organic and inorganic nanomaterials.

### 3.3. Discussion

In summary, titanium-based sonosensitizers, particularly TiO_2_ nanoparticles (NPs), have been extensively studied as inorganic sonosensitizers. However, the relatively low ROS generation efficiency, poor water dispersibility, and significant hindrance in tumor targeting severely impede its further application in the field of sonodynamic therapy (SDT). Manganese-based materials can selectively regulate the levels of O_2_ and GSH in the tumor microenvironment (TME), increase the production of ROS, and enhance the therapeutic efficacy of SDT. It is worth noting that further research is needed on the induction of immunogenic cell death (ICD) by manganese-based sonosensitizers. Although carbon-based materials can generate ROS under US irradiation, their inherent low charge separation efficiency and poor solubility limit the therapeutic efficacy of SDT. Further development of carbon-based materials can be pursued in the following aspects: utilizing specific ligands that demonstrate targeting capabilities to achieve targeted therapy for a certain type of cancer cells, introducing metal ions to regulate their charge transfer forms and utilizing their large surface area for drug loading. Silicon-based materials can also be enhanced for SDT through modification with noble metals or biopolymers.

For inorganic sonosensitizers, a potential direction for further development is the construction of nano therapeutic platforms, such as enhancing their dispersibility and stability in the matrix by utilizing other biocompatible polymers, improving their targeting ability through targeted modification materials or TEM-responsive materials, and constructing heterojunctions to suppress electron-hole recombination and thereby enhance ROS production.

It is worth noting that due to the diversity of materials and structures of inorganic sonosensitizers, molecular imaging can be utilized for guidance and monitoring of therapeutic processes in most SDT treatments. This is also one of the directions for future development. Current research has explored molecular imaging-guided SDT in several areas, such as magnetic resonance imaging (MRI) [99], ultrasound imaging [100], photoacoustic imaging [101], and fluorescence imaging [102]. Further development of novel inorganic sonosensitizers with imaging capabilities can be pursued to identify the optimal treatment window, meeting the current demands for precision and personalized therapy, and significantly improving treatment efficacy.

## 4. Sonodynamic Combination Therapy

Multiple dysregulations occur at various levels during tumorigenesis [103]. Thus, to enhance the therapeutic effect of cancer treatment, a multi-level approach must be taken to achieve synergistic therapy. Although sonodynamic therapy (SDT) is a promising treatment option, most sonosensitizers are inefficient in producing reactive oxygen species (ROS), and the tumor microenvironment (TME) further limits the efficacy of this approach. Recent studies have shown that combining SDT with phototherapy, chemodynamic therapy (CDT), chemotherapy, and immunotherapy can improve the overall treatment efficacy [104].

### 4.1. Combination of Phototherapy and SDT

During photothermal therapy (PTT), non-toxic photothermal agents are activated by irradiation at specific wavelengths, generating vibrational heat that triggers cell death in targeted tissues [105]. Combining photothermal therapy and sonodynamic therapy has shown high feasibility and effectiveness by increasing blood flow, promoting reactive oxygen species generation [106]. Photothermal therapy can induce reactive oxygen species production, and sonodynamic therapy can compensate for the limited light penetration, thereby eliminating tumors in deep tissues [107]. Nanomaterials with significant photothermal and acoustic dynamic effects are key to realizing SDT-PTT. To date, several nanoparticles with composite structures have been developed for anticancer treatment through SDT-PTT [108].

In biomedicine, titanium nitride (TiN) nanoparticles (NPs) are widely used due to their biocompatibility and physicochemical properties, including photothermal properties. TiN NPs contain tetravalent Ti and trivalent N bonds, leading to electron surpluses and vacancies in their lattice structure. In the case of microwave excited thermoacoustic therapy, an uneven distribution of electrons surrounds the vacancy, resulting in deficient electric dipoles. Wang et al. successfully synthesized TiN nanodots for cancer photothermal-enhanced SDT by liquid stripping [109]. Resulting TiN nanodots have satisfactory NIR II absorption properties and can be used to image tumors and treat them through photothermal therapy (PTT). It is worth noting that in the process of photothermal treatment, the surface of TiN nanodots will be partially oxidized to TiO_2_, so it can also be excited by ultrasound to generate ROS, which has the potential to be used as a sonosensitizer to achieve SDT. The mild photothermal heating caused by TiN nanodots under NIR II laser irradiation can optimize tumor blood flow and enhance tumor oxygenation, achieving a significant synergistic therapeutic effect through the combination of PTT and SDT.

Emerging therapies, such as sonodynamic and photothermal therapies, based on biocompatible black phosphorus (BP) nanomaterials have attracted attention due to their satisfactory synergistic effects, low side effects, and good biocompatibility. However, the sonodynamic treatment of pure black phosphorus nanosheets (BPS) has limited effectiveness due to rapid electron recombination resulting from their unique bandgaps. To solve this challenge, the modified nanocomposites (NCs) integrating gold nanoparticles and polypyrrole (PPy) with BPS were harvested through a moderate approach [110]. Through the enhanced permeability and retention effect, the prepared nanocomposites are accumulated into the tumor location. Eventually, integrating sonodynamic therapy of BPS and the photothermal effect of polypyrrole could lead to the satisfactory tumor inhibition effect. The nanocomposites showed excellent in vitro acoustic dynamics and photothermal conversion through tumor cells. Furthermore, Au-BPS-PPy-PEG nanocomposites have synergistic therapeutic effects and negligible side effects compared with most reported two-dimensional nanomaterials, which are attributed to enhanced electron transfer capability, good biocompatibility of BPS, and biocompatibility assessment, respectively. And PPy has significant photothermal conversion ability.

### 4.2. Combination of CDT and SDT

The CDT system is a tumor-specific therapeutic method that exploits the high levels of H_2_O_2_ and acidity of the TME to in situ produce highly toxic •OH via Fenton reactions [111]. Therefore, CDT and SDT are particularly effective for the treatment of deep-seated tumors.

Vanadium disulfide (VS2) has shown excellent performance in multimodal imaging and tumor therapy and has shown superior ability in biodegradation [112]. Inspired by previous studies, Lei et al. [113] constructed Fe-doped vanadium disulfide nanosheets (Fe-VS_2_ NSs) and further modified them with polyethylene glycol (PEG). According to the fluorescence intensity of the DCFH-DA staining experiment (Figure 8a,b), nanomaterials and H_2_O_2_ incubation group and nanomaterials and US irradiation group were observed to emit a certain amount of green fluorescent signal, indicating that ROS was produced. The nanomaterials group incubated with H_2_O_2_ and irradiated with ultrasonic can be observed to emit a strong fluorescent signal, indicating that a large amount of ROS was produced. In conclusion, SDT and CDT combined treatment significantly increased the killing effect of tumor cells compared with single treatment. Furthermore, the green fluorescence signal of ThiolTracker Violet decreased significantly with increasing Fe-VS_2_-PEG concentration (Figure 8c). The weaker the green signal, the lower the glutathione content. Fe-VS_2_-PEG NSs were found to be effective at removing intracellular GSH. In conclusion, the killing mechanism is shown in Figure 8d. Firstly, Fe-VS_2_-PEG NSs can utilize endogenous H_2_O_2_ in the tumor microenvironment for chemokinetic therapy (CDT). Secondly, as a sound-sensitive agent, ROS can be generated under the condition of ultrasound. Finally, the polyvalent iron and vanadium elements in Fe-VS_2_-PEG NSs can deplete glutathione, thereby amplifying oxidative stress induced by reactive oxygen species via SDT and CDT.

Notably, Cu^2+^ or Mn^2+^-mediated Fenton-like reactions, which also convert active H_2_O_2_ to cytotoxic hydroxyl radical (•OH), enable CDT [114]. Gong et al. [115] reported a novel Cu-CuFe_2_O_4_ nanoenzyme capable of simultaneously relieving hypoxia and depleting glutathione, thus effectively enhancing ROS-involved chemical kinetics (CDT) acoustic kinetics (SDT) therapy. The nanoenzyme has catalase-like and glutathione peroxise-like catalytic activities and can continuously catalyze the formation of oxygen (O_2_) from tumor overexpressed hydrogen peroxide (H_2_O_2_) to facilitate the formation of ^1^O_2_ under external sonication for hypoxia relief SDT. Meanwhile, Cu-CuFe_2_O_4_ NP reacts with GSH, depleting GSH and releasing Fenton-like Cu^+^ and Fe^2+^ ions to mediate the production of a large number of hydroxyl radicals (•OH) in CDT. It was proved that Cu-CuFe_2_O_4_ NP had a significant synergistic therapeutic effect of SDT and CDT by efficient killing of MCF-7 cells.

### 4.3. Combination of Chemotherapy and SDT

Due to the high toxicity of chemotherapy drugs, chemotherapy is considered the most effective cancer treatment today. Ultrasound irradiation can cause damage to the cell membrane, allowing more drugs to accumulate inside the cell [116]. Additionally, many inorganic sonosensitizers can be cleaved under the action of ultrasound, releasing the internal drugs they carry [117].

The chemotherapy drug doxorubicin (DOX) treats breast cancer by forcing DNA strand breaks that induce apoptosis. Notably, doxorubicin itself also has the ability to be stimulated by US to release ROS. However, DOX itself has limited bioavailability and low tumor accumulation, which is easy to cause damage to systemic cells [118]. Liang et al. [119] designed Pt-doped TiO_2_ nanoparticles with hollow cavities for doxorubicin loading (HPT–DOX). The synthesis process and DOX loading are shown in the Figure 9a. Under US radiation, DOX loaded in a TiO_2_ cavity can act both as a chemotherapy drug and as a molecular sonosensitizer. In this way, the damage to normal cells caused by conventional chemotherapy is effectively reduced.

Turmeric’s rhizome is a natural source of the polyphenol Cur, which exhibits a range of pharmacological properties, including inducing cancer cell apoptosis and inhibiting tumor metastasis, invasion, and angiogenesis [120,121]. Cur combined with US irradiation demonstrated significantly enhanced inhibitory effects on various cancer cells and bacteria [122]. Tian et al. [123] synthesized gadolinium doped hollow silica nanospheres (GD-HMSN) and used them as nanocarriers for Cur loading (Figure 9b). When triggered by a weakly acidic microenvironment and US irradiation, Cur@Gd-HMSNs-CMD can be degraded. In this way, loaded Cur can be released, allowing T1 contrast to be enhanced. The good synergistic effect of the sonodynamic force and chemotherapy produced a satisfactory therapeutic effect in the 4T1 tumor model.

### 4.4. Combination of Immunotherapy and SDT

The combination of immunotherapy and SDT has gained recent attention as a potential strategy for controlling tumor growth and inducing antitumor immunity. T-cell effector functions are often enhanced as part of cancer immunotherapy [124]. PD-L1 antibodies, for instance, block the binding between tumor cells and their cognate receptor, PD-1. Blocking anti-PD-L1 facilitates antitumor immunity by activating tumor-specific T cells in a targeted manner. PD-L1 inhibitors have recently bound to HMME-directed SDT [125], which enhances the production of dendritic cells and proinflammatory IL-6 and TNF-α cytokines in tumors. This study indicates that anti-PD-L1 + SDT is effective in reducing both tumor volume and metastatic lung nodules, as well as increasing CD8+T effector cells, suggesting the combination of anti-PD-L1 + SDT can have a significant antitumor effect.

Specific antigens expressed on cancer cell membrane-coated nanoparticles can be processed by mature antigen-presenting cells to trigger anticancer immunity based on cytotoxic T lymphocytes. Antibodies against programmed cell death ligand 1 (aPD-L1) bind specifically to PD-L1 expressed on tumor cells, acting as immune checkpoints in an immune environment for tumors. Thus, Wei et al. [126] reported a novel TiO_2_-based sonosensitizer for dual targeting to enhance SDT (Figure 9c). Under US irradiation, the synthesized sonosensitizer can catalyze the formation of a large amount of ^1^O_2_. In vitro experiments showed that the functionalized sonosensitizers had a precise targeting effect, high uptake rate of tumor cells, and intracellular acoustic catalytic killing of B16F10 cells by a large number of local ROS. Using animal models of melanoma, functionalized SCNS show visible long-term retention in tumor regions, which contributes to tumor homology and synergistic double targeting of immune checkpoints to enhance SDT in vivo.

**Figure 9 ijms-24-12001-f009:**
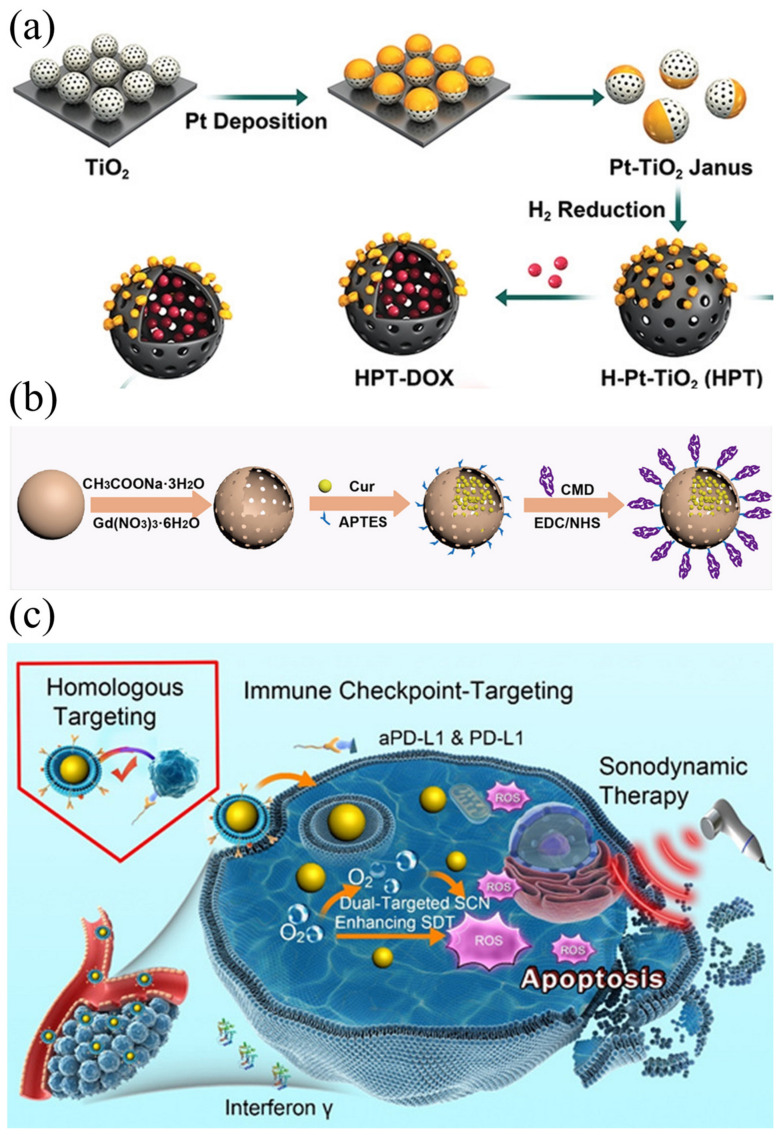
(**a**) Schematic illustration for the preparation of HPT–DOX [119]; (**b**) Cur@Gd-HMSNs-CMD nanocomposite designed for combined SDT and chemotherapy [123]; (**c**) schematic representation of SCN@B16F10M/PEG-aPD-L1 dual targeting of homology and immune checkpoints to tumors and enhanced SDT [126].

Table 1 summarizes representative SDT-based combination therapy.

### 4.5. Discussion

Combination therapy offers advantages over monotherapy, as it not only improves the tumor microenvironment, but also works synergistically to achieve optimal therapeutic results. For example, (1) combined with SDT and PTT, under the influence of the photothermal effect, the therapeutic effect will also be improved. (2) CDT used in combination with SDT can selectively induce ROS in tumor cells through a Fenton-like reaction, and ultrasound enhances this reaction by generating heat. (3) Combining SDT with chemotherapy may enhance tumor cell sensitivity to chemotherapy drugs by downregulating ATP-binding cassette transporters. However, there are still challenges to overcome in combination therapy. Multi-component inorganic nanomaterials may produce unexpected biotoxicity or be difficult to degrade. Additionally, conflicts can arise in combination therapies based on SDT, such as inhibition when SDT and PDT both consume oxygen. Addressing these issues is essential for future progress in the field.

Clinical translation is the bridge between basic research and clinical practice, serving as a crucial component in the transformation of scientific research findings into clinical applications. Currently, the majority of cases reported are SDT combined therapy [140,141]. In most cases, SDT has demonstrated excellent therapeutic efficacy, resulting in positive clinical outcomes. Additionally, only mild adverse effects have been reported, primarily involving minor pain at the site of bone metastasis.

## 5. Conclusions and Future Directions

Nanotechnology and nanomedicine have undergone significant advancements in the past few decades, resulting in rapid interdisciplinary developments in the integration of nanotechnology and medicine. Inorganic nanoparticles are stable, non-phototoxic, and possess a longer blood circulation time compared to organic nanoparticles, making them an extensively studied option for cancer SDT. However, to enhance the efficacy of inorganic materials as sound-sensitive agents, we require a comprehensive understanding of their physical properties and analysis of the mechanism underlying SDT-induced tumor cell death. The main purpose of this review is to systematically review the mechanism and application of inorganic nanomaterials in SDT, and provide inspiration for the screening and further exploration of inorganic sonosensitizers in the future. Despite significant efforts being made to develop highly effective ultrasound sensitizers, none have yet been able to enter clinical treatment. Therefore, further efforts are required to address key issues before SDT can undergo clinical translation.

First of all, achieving the translation of SDT from bench to bedside requires significant attention to the biosafety of inorganic nanoparticles. This involves the assessment of intracellular uptake, localization, biodegradation, excretion, and systemic toxicity. Inorganic–organic hybridization allows for the biodegradation of inorganic nanoparticles by introducing biodegradable organic components. Although some studies have been carried out on the biosafety of SDT, a comprehensive evaluation of the biological safety of inorganic materials and corresponding research is still needed for clinical applications. Therefore, it is necessary to investigate the biosafety mechanisms of inorganic nanoparticles.

Secondly, the development of clinically useful SDT devices with tunable acoustic parameters and accurate acoustic location is required. A complete sound power unit must include probe design, ultrasonic driving units, echo signal processing, display, and control units. In clinical applications, it is also necessary to consider the interaction with the patient and achieve personalized and precise treatment. It should not be ignored that there is no uniform standard for the parameters used in the existing studies, and it is difficult to determine the difference in the anti-tumor efficiency of different sonosensitizers. Therefore, it is also imperative to develop appropriate standards for the future development of SDT devices.

Finally, the current mechanism of SDT is not clear enough, and the relationship between the physicochemical properties of acoustic sensitizers and the therapeutic effect cannot be quantified, which brings obstacles to the design of acoustic sensitizers with excellent SDT properties. At present, it is unknown whether cavitation or ROS production plays a dominant role in the SDT process. The specific process of ROS production in tumors has not been clarified. These problems need to be further studied by new experimental methods and advanced characterization techniques.

While previous studies have demonstrated the superiority of SDT, the clinical application of sonosensitizers requires a lengthy development process. However, as sonosensitizers continue to advance, they may find application in a variety of biomedical fields in the future.

## Figures and Tables

**Figure 1 ijms-24-12001-f001:**
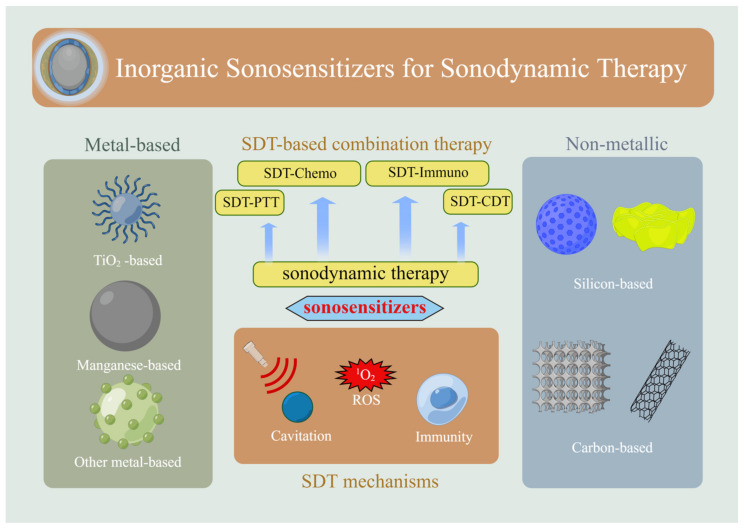
Overview of the research on inorganic sonosensitizers.

**Figure 2 ijms-24-12001-f002:**
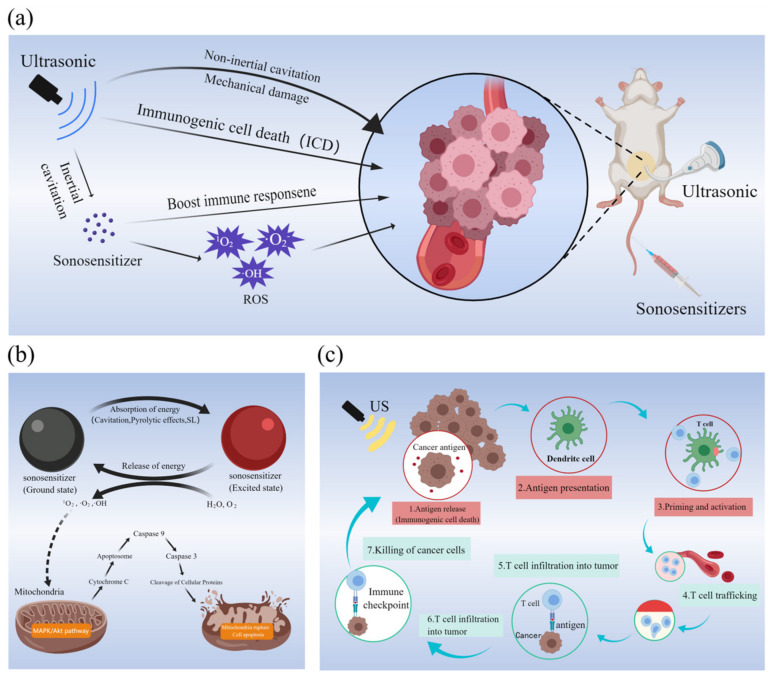
(**a**) Possible mechanisms of tumor killing by SDT; (**b**) mechanism of ROS production and the process of ROS killing mitochondria; (**c**) schematic diagram of immunotherapy process induced by SDT. (**a**–**c**) Created with MedPeer (www.medpeer.cn, accessed on 25 September 2022).

**Figure 3 ijms-24-12001-f003:**
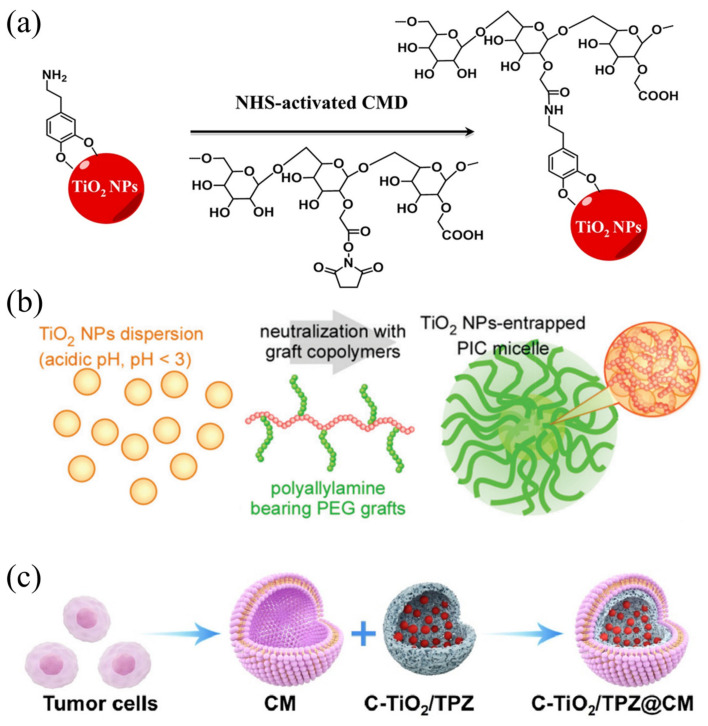
(**a**) Surface modification of TiO_2_ NPs [53]; (**b**) schematic image of TiO_2_ NP-entrapped polyionic composite micelles [54]; (**c**) schematic diagram of surface modification of C-TiO_2_/TPZ@CM nanoparticles [58].

**Figure 4 ijms-24-12001-f004:**
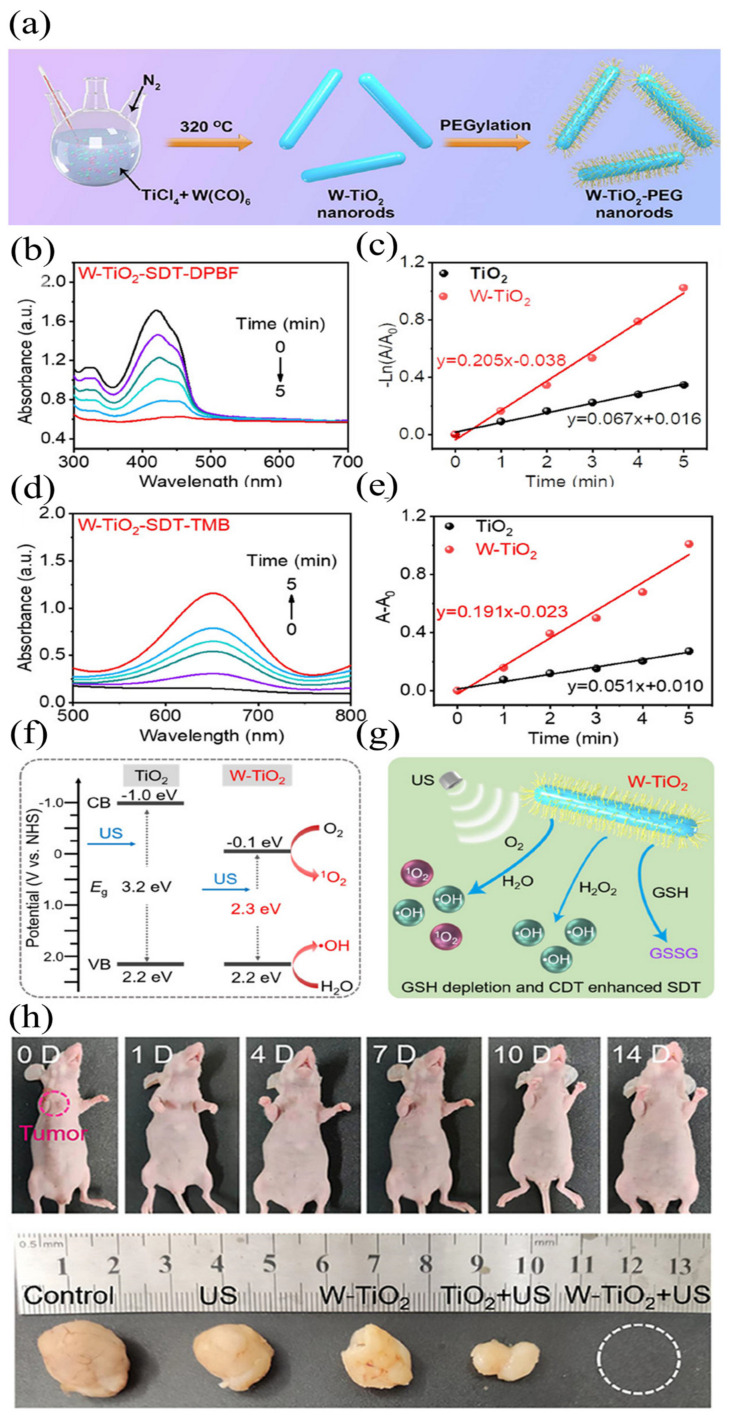
(**a**) Preparation of W-doped nanocomposites; (**b**,**c**) time-dependent ^1^O_2_ (**b**) and •OH (**c**) generation of nanocomposites; (**d**,**e**) Rate constant of ^1^O_2_ (**d**) and •OH (**e**) generation in the presence of TiO_2_ or W−TiO_2_ nanorods; (**f**) Nanocomposites and TiO_2_ nanorods energy-band diagrams; (**g**) a schematic illustration of enhanced ROS generation using W−TiO_2_ nanorods to deplete GSH and form combination therapy; (**h**) photographs of mice treated with CDT-enhanced SDT and their tumors [60]. The lines in (**b**) mean the ultrasound irradiation time (from top to bottom: 0, 1, 2, 3, 4, 5 min), and for (**d**), the lines mean the ultrasound irradiation time (from top to bottom: 5, 4, 3, 2, 1, 0 min).

**Figure 6 ijms-24-12001-f006:**
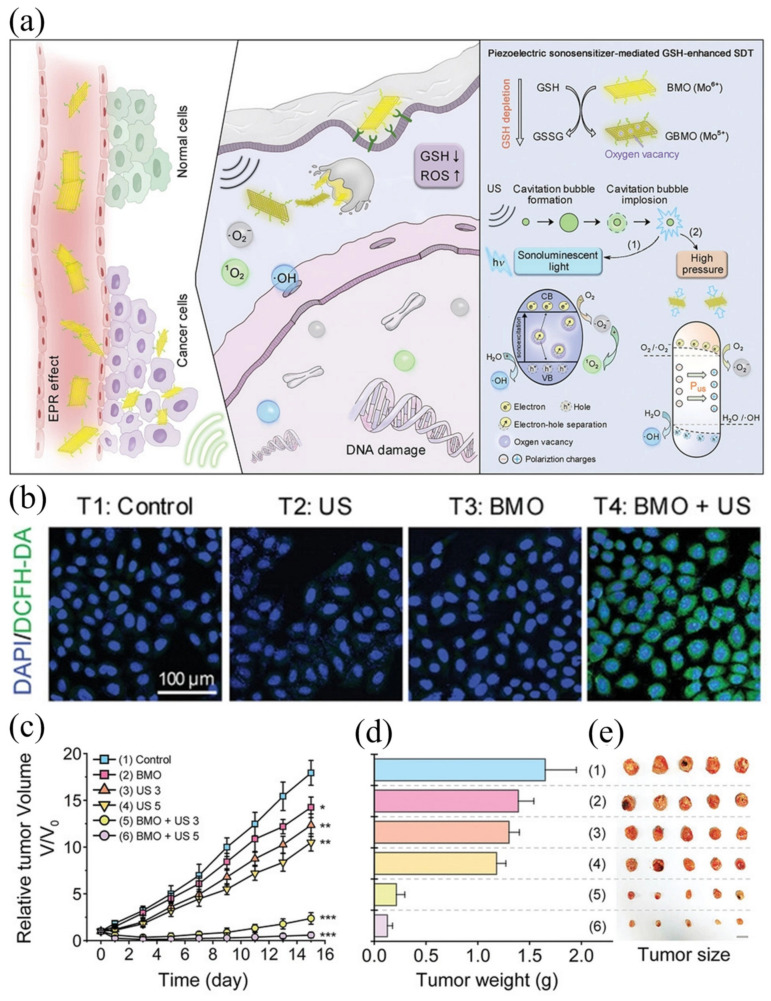
(**a**) Schematic diagram of the 2D piezoelectric Bi_2_MoO_6_ sonosensitizer for GSH−enhanced sonodynamic therapy; (**b**) intracellular ROS level with BMO NRs for 4 h followed by various treatments; (**c**) relative tumor growth curves of tumor-bearing mice after different treatments; (**d**) the average weights; and (**e**) photo of tumors dissected from the representative mice 15 d after various treatments [77]. *, ** and *** in this figure means: * *p* < 0.01, ** *p* < 0.005, *** *p* < 0.001.

**Figure 7 ijms-24-12001-f007:**
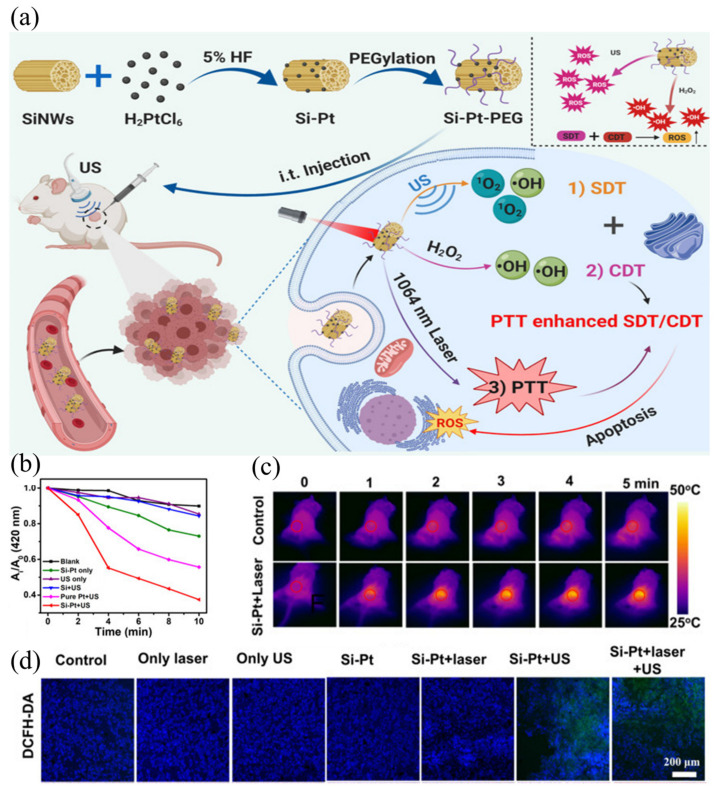
(**a**) Schematic illustration of a nanocomposite that can be excited by light and ultrasound to achieve synergistic SDT therapy; (**b**) DPBF oxidation by Si-Pt only, SiNWs only, and pure Pt under US irradiation for 10 min; (**c**) IR images after intratumoral injection of Si-Pt NCs under the laser irradiation; (**d**) fluorescence images of DCFH-DA stained tumor slices collected from mice 24 h post treatment [84].

**Figure 8 ijms-24-12001-f008:**
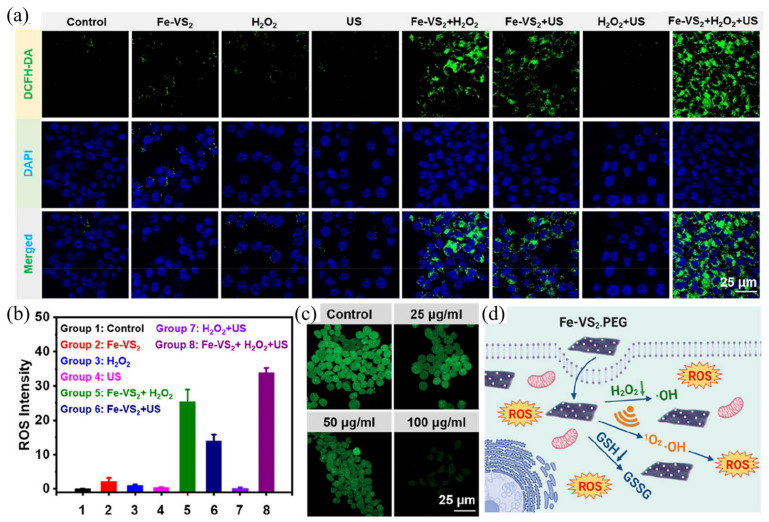
(**a**) Laser scanning confocal microscope images of ROS production in different control groups; (**b**) intracellular levels of reactive oxygen species production; (**c**) laser scanning confocal microscope images of tumor cells and nanocomposites stained after incubation; (**d**) cell killing mechanism of nanocomposites [113].

**Table 1 ijms-24-12001-t001:** SDT-based combination therapy.

Sonosensitizer	Treatment Modalities	US Parameters	Target	Ref.
Au NPL@TiO_2_	SDT + PTT	3.0 MHz 0.5 W/cm^2^	HeLa cells	[127]
Ir-B-TiO_2_@CCM	SDT + PTT	1.0 MHz 1 Wcm^2^	HeLa cells	[128]
ZrO_2−x_@PEG	SDT + PTT	1.0 MHz 0.5 W/cm^2^	4T1 cells	[129]
Porous carbon nanospheres	SDT + PTT	1.0 MHz 1 W/cm^2^	4T1 cells	[130]
Au-MnO Nanoparticles	SDT + CDT	1.0 MHz 2 W/cm^2^	MCF-7cells	[131]
CoFe_2_O_4_	SDT + CDT	1.0 MHz 1 W/cm^2^	4T1 cells	[132]
Fe-TiO_2_ nanodots	SDT + CDT	40 kHz 3 W/cm^2^	4T1 cells	[133]
AIPH@Cu-MOF	SDT + CDT	1.0 MHz 0.5 W/cm^2^	Panc02-Luc cells	[134]
Au-PTX NPs	SDT + chemotherapy	1.0 MHz 1 W/cm^2^	C540 cells	[135]
Dox@FeCPs	SDT + chemotherapy	1.0 MHz 1.75 W/cm^2^	CT26 cells	[136]
TiO_2_: Gd@DOX/FA	SDT + chemotherapy	3.3 MHz 1 W/cm^2^	LNCaP cells	[137]
TiO_2_-Ce6-CpG	SDT + immunotherapy	1.0 MHz 1 W/cm^2^	Hepa1-6	[138]
(TPP)/R837@M	SDT + immunotherapy	3.0 MHz 1.5 W/cm^2^	4T1 cells	[139]

## Data Availability

Not applicable.

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
