# Peer review of "A Comprehensive Review of Inorganic Sonosensitizers for Sonodynamic Therapy"

_ijms, 2023, doi:10.3390/ijms241512001_

Round 1

Reviewer 1 Report

The manuscript presents selected achievements in field of sonosensitizers for sonodynamic therapy. The article provides relevant information to the field nevertheless, some aspects could be further clarified to increase its impact.

1. Can the presented sonosensitizers be useful in diagnostics and be the basis of the theranostics platform? What imaging modalities can sonosensitizers help with?

2.  Is there also a hyperthermic effect when using ultrasound? What is temperature rises role?

3. Only selected nanoparticles used as sonosensitizers are presented. There is no description of the use of another inorganic materials, e.g. iron oxide nanoparticle mediators.

4. There is not detailed, critical discussion about the result and perspective. There is no discussion about which particles are the most promising and worth researching (roadmap for other scientists).  

Reviewer 2 Report

Although the chosen topic is of great interest, the authors do not have a proper structure for this research.

The abstract does not follow the guidelines for author’s structure. Most of the abstract is about the introduction.

The title does not show the type of the article.

 Line 100, the authors mention that this paper is a review but do not specify what type and do not follow the proper structure of a review article.

Line 100, “this review aims to speculate”, in a scientific paper the reader must found sound scientific arguments for the techniques and methods presented, not speculations.

The authors should identify the real aim and secondary objectives of the paper and using an appropriate methodology to demonstrate these things and to conclude accordingly at the end of the study.

Lines 106- 108 “this article  introduces a method to enhance the killing effect on tumor cells, which involves combining sonodynamic therapy with other existing therapies to make up for their respective deficiencies and achieve mutually reinforcing effects”. Usually a literature review does not introduce a method, but rather comes with solid information about its efficiency in accordance with the findings of other reliable studies, at the same time doing an analysis of their degree of bias.

Most of the figures have bibliographic reference, do you have the consent of their authors for their use in this paper?

Lines 753 – 754: The conclusion “While previous studies have demonstrated the superiority of SDT, the clinical application of sonosensitizers requires a lengthy development process.” Is not supported by sound data analysis.

no comments about the quality of English Language 

Round 2

Reviewer 1 Report

The manuscript was improved and may be published.